# Adaptive foraging of pollinators fosters gradual tipping under resource competition and rapid environmental change

Sjoerd Terpstra[1,2,3], Flávia M. D. Marquitti[4,5], Vítor V. Vasconcelos[3,6] *

**1** Graduate School of Informatics, University of Amsterdam, Amsterdam, The Netherlands, **2** Institute for Marine and Atmospheric Research Utrecht, Utrecht University, Utrecht, The Netherlands, **3** Institute for Advanced Study, University of Amsterdam, Amsterdam, The Netherlands, **4** Instituto de Física 'Gleb Wataghin' & Programa de Pós Graduação em Ecologia, Universidade Estadual de Campinas, Campinas, São Paulo, Brazil, **5** International Centre for Theoretical Physics - South American Institute for Fundamental Research (ICTP-SAIFR), São Paulo, São Paulo, Brazil, **6** Computational Science Lab, Informatics Institute, University of Amsterdam, Amsterdam, The Netherlands

\* v.v.vasconcelos@uva.nl

**Data Availability Statement:** The data and model are published online as: Terpstra, Sjoerd, Marquitti, Flávia M.D., & Vasconcelos, Vítor V. (2023). Model Implementation of: Adaptive Foraging of Pollinators

## Abstract

Plant and pollinator communities are vital for transnational food chains. Like many natural systems, they are affected by global change: rapidly deteriorating conditions threaten their numbers. Previous theoretical studies identified the potential for community-wide collapse above critical levels of environmental stressors—so-called bifurcation-induced tipping points. Fortunately, even as conditions deteriorate, individuals have some adaptive capacity, potentially increasing the boundary for a safe operating space where changes in ecological processes are reversible. Our study considers this adaptive capacity of pollinators to resource availability and identifies a new threat to disturbed pollinator communities. We model the adaptive foraging of pollinators in changing environments. Pollinator's adaptive foraging alters the dynamical responses of species, to the advantage of some—typically generalists—and the disadvantage of others, with systematic non-linear and non-monotonic effects on the abundance of particular species. We show that, in addition to the extent of environmental stress, the pace of change of environmental stress can also lead to the early collapse of both adaptive and nonadaptive pollinator communities. Specifically, perturbed communities exhibit rate-induced tipping points at stress levels within the safe boundary defined for constant stressors. With adaptive foraging, tipping is a more asynchronous collapse of species compared to nonadaptive pollinator communities, meaning that not all pollinator species reach a tipping event simultaneously. These results suggest that it is essential to consider the adaptive capacity of pollinator communities for monitoring and conservation. Both the extent and the rate of stress change relative to the ability of communities to recover are critical environmental boundaries.

## Author summary

Plant and pollinator communities, which support global food chains, are threatened. A significant problem is the reduction of pollination, where not enough plants are pollinated

Fosters Gradual Tipping under Resource Competition and Rapid Environmental Change (Zenodo). Zenodo. https://zenodo.org/doi/10.5281/zenodo.8268437.

**Funding:** ST acknowledges the support of the Institute of Advanced Study (University of Amsterdam, IAS excellence students 2022) where part of the work was carried out. FMDM acknowledges the Coordenação de Aperfeiçoamento de Pessoal de Nível Superior Brazil (Finance Code 001). VVV declares financial support of the CSL at IvI (Computational Science Lab at Informatics Institute at the University of Amsterdam, Tenure Track). The funders had no role in study design, data collection and analysis, decision to publish, or preparation of the manuscript.

**Competing interests:** The authors have declared that no competing interests exist.

and are at risk of becoming extinct. Environmental change, such as climate change or increasing pesticide use, can cause this reduction. As long as these changes stay under certain boundaries, undesirable ecological processes like the reduction in pollination are reversible. Crossing these boundaries means tipping of pollinator communities might occur; the system might collapse. Pollinator communities can adapt, for instance, through adaptive foraging by changing which plants they prefer in response to environmental deterioration. We include adaptive foraging in a theoretical model of plant and pollinator communities. We show that plant and pollinator communities are vulnerable to the extent of stressors but also to how fast these stressors increase. Our model shows that, with adaptation, the extinction of species is more sequential and spreads out in time and may even temporarily favor especially generalist species.

## Introduction

Pollinator communities, comprising interconnected plant and pollinator species, are vital for many ecosystems and enhance their biodiversity [1]. Interactions at the level of these communities ensure the circulation of nutrients. Pollinators pollinate an estimated 87.5% of all existing flowering plant species [2], some of which are the only ones supplying certain vital nutrients to humans [3, 4]. Furthermore, the food industry is highly dependent on flowering plant species, with approximately 75% of crop species and 35% of crop volume relying on pollinators [5]. These numbers show the importance of the mutualistic relationship between plants and pollinators to humans and the biosphere.

Plants provide food to pollinators and pollinators promote pollen grains transference between individual plants, but the exact causal mechanisms that ensure the persistence of pollinator communities remain unclear. Still, one can represent how species interact with each other using bipartite networks, where one set of nodes represents pollinator species and the other set of nodes plant species. Typically, these networks that describe mutualistic interactions consist of a core of generalist species—species that have numerous interactions, many of which they share with other generalists [6, 7]—and a peripheral group of specialists—species with only one or two interaction partners— mostly attached to the generalists. This structure is called a nested network [6, 8, 9]. Nested networks are highly asymmetric since specialists primarily depend on generalists, whereas generalists are less dependent on each of their specialists [6, 7]. Nestedness can help rare specialist species persist [6].

Understanding how pollinator communities remain healthy is crucial, given the declining populations of many species [1, 3, 10, 11]. Multiple environmental stressors contribute to this decline in pollinators. The IPBES report on pollinator trends specifies land-use change, intensive agricultural management and pesticide use, environmental pollution, invasive alien species, pathogens, and climate change as possible drivers of pollinator decline [12]. A panel of 20 experts in the field of pollinator studies ranked the most important drivers of decline globally, in decreasing order of impact, as land cover and configuration, land management, pesticide use, climate change, pests and pathogens, pollinator management, invasive alien species, and genetically modified organisms [13].

Some of these drivers of decline have changed rapidly over the last decades. For instance, rapid climate change has likely been decreasing pollinator diversity in the Northern Hemisphere, leading to a loss of resilience of these pollinators and their ecosystems [14]. Furthermore, a changing climate can cause plants and pollinators to shift out of their climate range or mismatch their phenology and active hours, leading to the decline and possible extinction of

species [3]. Moreover, climate change can lead to species migration and the invasion of other pollinator communities, which can lead to a loss of diversity in those areas [3, 7].

The impact of external stressors on pollinator communities may have far-reaching and non-obvious consequences on ecosystems. For instance, nonlinear dynamics, which can be captured in models of pollinator communities, can present tipping points. Tipping points are defined as sudden, abrupt changes in a system due to a small change in a control parameter [15–20]. This control parameter induces a transition from one state of the system where the community is prosperous to a depauperate one at some critical threshold. Planetary boundaries are the ultimate example of these thresholds [21] above which there is a chance of long-term devastating consequences such as large, irreversible sea level rise [22].

So far, there is no direct empirical evidence for tipping in pollinator communities [23]. In their review, [23] found just one example of tipping in a bee colony [23, 24]. However, some of the theoretical work on pollinator communities has shown that tipping points are possible, especially in models based on Lotka-Volterra type of dynamics [23, 25–29]. Various theoretical works tested the existence of different equilibrium states of the communities for different but fixed values of the control parameters (e.g., interaction strength or driver of decline) [25–28]. Results show that higher drivers of decline cause community collapse [26]. This type of tipping is formally associated with the bifurcation of the equilibrium solutions. Bifurcation-induced tipping is caused by reaching a threshold value of a control parameter—here, the magnitude of the driver of decline—and that critical value is traditionally seen as the "tipping point". However, since these works investigated equilibrium solutions at each subsequent value of the driver of decline (quasi-static analysis), they did not consider the complex time-dependent dynamics that can arise by varying a parameter over time [30]. The equilibrium state might actually be a moving quasi-static attractor, making it possible for rate-induced tipping to exist [31]. Rate-induced tipping occurs due to a rate of change in the control parameter instead of due to reaching a threshold value of the parameter. Conversely, what matters in bifurcation-induced tipping is the absolute value of the parameter, not how slowly or rapidly that value was reached. By ignoring that control parameters are time-dependent, changes in dynamics due to the rate of change are overlooked [19], potentially overestimating the resilience of such systems. Further, it disassociates the critical values of external environments (tipping points) from any particular point in time. Nevertheless, the possibility and risk of rate-induced tipping through rate dependency have so far been ignored in the study of pollinator communities.

On the positive side, the ability to adapt could give pollinator communities greater resilience and, thus, is expected to play an important role in rate-induced tipping. Adaptation, however, is not instantaneous. For adaptation to be an effective coping mechanism, it must occur on a timescale similar to the pace at which the environment changes. Therefore, environmental change should be seen in light of the rate of adaptation. Rate dependence and adaptation have been underappreciated in previous analyses of tipping points in pollinator communities, even though they would bring models closer to the dynamics observed in nature. When the environment changes rapidly, plasticity and evolution are important mechanisms for adaptability.

There is evidence that pollinators show behavioral plasticity in their use of resources and employ various strategies, such as changing the foraging intensity of different species (adaptive foraging) and foraging different species altogether (rewiring). Pollinator communities experience high species turnover and fluctuating abundances [32]. This results in high interaction variability, which can be modeled by the opportunistic attachment of pollinators to plants [33], potentially forming novel or reinforcing existing interactions. By changing their connections, species can adapt quickly to changes in their environment [34]. A sign that this is happening is a high turnover rate of interaction between species [35]. However, this stems mainly

from the variability in the number of active links per species rather than an actual rewiring between species (i.e., forming connections between species that were not previously connected) [35]. Here, we represent these mechanisms in a new model of adaptive foraging.

With adaptive foraging, pollinator species can adapt their investment in plant species to which they can connect, thereby altering the network weights. This strategy is based on the availability of plant resources, which depends on plant abundance and competition between pollinators. Adaptive networks seem to improve the persistence of pollinator communities, and adaptive foraging sustains biodiversity in pollinator communities [36].

As environments keep changing, adaptation may play an increasingly important role. Therefore, including adaptation and dependencies on the rate of change of environmental stressors in current models of pollinator communities is necessary to investigate whether rate-induced tipping is present and how it happens. In this work, the questions are whether i) in adaptive pollinator communities, irreversible tipping can occur due to the speed of environmental change, not just due to the extent of environmental change, and ii) how adaptive foraging changes the overall tipping dynamics of the model. The focus is on adaptive mechanisms at the ecological timescale, not the evolutionary one, since the latter acts at longer timescales. We will use the term adaptation in this sense. We expand on previous models [26] by adding adaptive foraging depending on resource competition and a rate dependency to the driver of decline.

## Model

The model is based on previously used mutualistic Lotka-Volterra type models describing pollinator communities (e.g., [26, 27, 29]), extended to include implicit resource competition and adaptive foraging. Fig 1A illustrates the causal mechanisms considered in our model.

### Abundance changes

The abundance of each plant ($P_i$) and pollinator species ($A_j$) is modeled considering three main components: intrinsic growth rate, mutualistic growth, and intraguild competition (see Fig 1), each contributing to net population growth. These components are described below and are reflected in the following coupled model of plant species indexed by $i$, in a set of $S^P$ plant species, and pollinator species indexed by $j$, in a set of $S^A$ pollinator species,

$$
\begin{aligned}
\frac{dP_i}{dt} &= P_i\left(r_i^P - \sum_{i'=1}^{S^P} C_{ii'}^P P_{i'} + H\left(\sum_{j'=1}^{S^A} \alpha_{ij'}\beta_{ij'}^P A_{j'}\right)\right) + \mu, \\
\frac{dA_j}{dt} &= A_j\left(r_j^A - d_A(t) - \sum_{j'=1}^{S^A} C_{jj'}^A A_{j'} + H\left(\sum_{i'=1}^{S^P} \alpha_{i'j}\beta_{i'j}^A \phi_{i'}\right)\right) + \mu.
\end{aligned}
\tag{1}
$$

**Intrinsic growth rate.** Each species has a baseline, fixed intrinsic growth rate given by $r_i^P$ and $r_j^A$ for plant and pollinator species, respectively.

**Driver of decline.** The total growth rate of pollinator species is negatively affected by the driver of decline $d_A(t)$, representing the total environmental deterioration. This driver of decline is either kept fixed or is time-dependent: $d_A(t) = \lambda \cdot t$ with $\lambda$ the rate of change.

**Intraguild competition.** The intraguild competition, where each species competes with all species in its own group (i.e., plant species compete with plant species, pollinator species compete with pollinator species), is a nonlinear term depending on the abundance of the

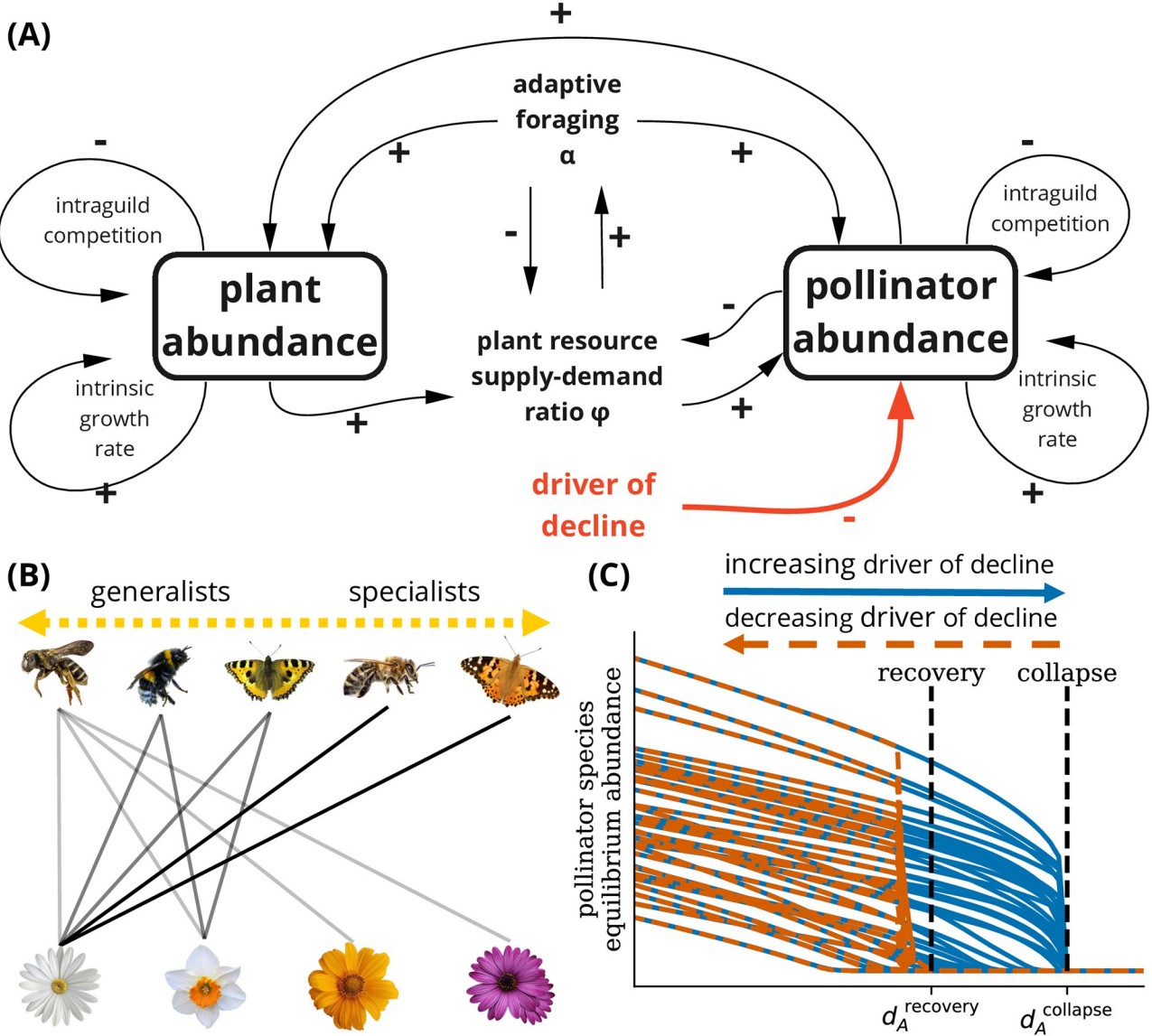

**Fig 1. Dynamics of pollinator communities under external stressors. (A)** Causal dynamics of the adaptive foraging model. Plants and pollinators boost each other's abundance through mutualistic interactions; pollinators receive resources from plants and plants are pollinated by pollinators. Interactions are specified by a bipartite plant-pollinator nested network. The strength of the interactions changes over time through adaptive foraging. Pollinators adapt their investment in connected plant species based on the supply-demand ratio of plant resources. Besides, species experience intraguild competition and an intrinsic growth rate. The system is subjected to a driver of decline ($d_A$)—an external stressor that negatively influences the intrinsic growth rate of all pollinator species. **(B)** The mutualistic network has a nested structure with a few highly connected species (generalists) and many sparsely connected species (specialists). The edge colors show the weights of the adaptive foraging matrix $\alpha$, denoting the investment of each pollinator species in plant species. **(C)** Hysteretic impact of driver of decline $d_A$ (without adaptive foraging). At a critical value of the driver of decline, for any increase of the driver of decline, the pollinator community undergoes bifurcation-induced tipping. At this tipping point, the pollinator community collapses—all pollinator species go extinct (very low abundance). By lowering the driver of decline, the system has another bifurcation-induced tipping point, where, for any decrease of the driver of decline, it recovers at the point of recovery where the first species are reintroduced.

species involved and competition matrices $C_{ii'}^{P}$ and $C_{jj'}^{A}$ for plant and pollinator species, respectively.

**Mutualistic interactions with resource competition.** A Holling type-II function gives the mutualism-derived growth, $H(\rho)$, which ensures the saturation of the mutualistic relation as the overall mutualistic benefit, $\rho$, increases. Precisely, $H(\rho) \equiv \rho(1 + \rho/h)^{-1}$, with $h$ as a

horizontal asymptote with a different value for each species [37]. The mutualistic benefits, $\rho$, are the sums passed to the $H$ functions for pollinators and plants in Eq 1. They are different for pollinators and plants, so we call them $\rho_j^A$ and $\rho_i^P$, respectively.

To establish the mutualistic benefit for plant and pollinator species, we extend previous models (e.g., [26, 27, 29]) by considering that pollinators can change their strength of interaction with each of their connected plant species. The strength of interaction and the resources available determine the mutualistic benefit the species derive.

Formally, this is captured by the element-wise product of two matrices, an evolving foraging effort $\alpha$ and a static matching matrix $\beta$, that, together with resource availability, make up the interaction strength.

The matching matrix $\beta$ encodes whether a connection between a plant and pollinator species can exist and its innate efficiency. The innate efficiency describes how well two species can interact with each other given their phenotype, the accessibility of the plant to the pollinator, and other unknown constraints [38]. Matrix $\beta$ also contains forbidden links (FL), connections that cannot exist due to phenological uncoupling [38]. We assume that evolutionary processes do not play a role at the timescales we consider, and no species can rewire their interactions completely; therefore, $\beta$ is fixed.

Whether two species can interact with each other (and, thus, whether the $\beta$-value between two species is nonzero) is dictated by a bipartite network describing the connections. We generate these bipartite networks using an algorithm as described in Section B of S1 Text. The algorithm takes as input the number of plant species, number of pollinator species, nestedness, fraction of forbidden links, and connectivity. These networks have a similar—nested—structure as real-life plant-pollinator networks [8, 26].

The foraging effort, $\alpha$, is the relative investment a pollinator species $j$ makes in a certain plant species $i$. It encompasses the effort—the amount of resources and time—that pollinators decide to invest in each plant. The pollinators, not the plants, are, therefore, leading in the choice of mutualistic interaction strengths. Plants are assumed to have no selective mechanism apart from the matching $\beta$ and their resource abundance. High matching and high abundance make plants more attractive to pollinators. The total foraging effort is constrained by $\sum_i^{S^P} \alpha_{ij} = 1$, meaning that each pollinator species $j$ divides a total foraging effort of 1 across all connected plant species. The foraging effort $\alpha$ is initialized such that the initial foraging effort is evenly distributed over all connected plant species.

The values for $\beta$ are sampled uniformly for each connection existing in the bipartite network and subsequently normalized by the number of connections a plant ($K_i^P$) or pollinator ($K_j^A$) species has. As a second step, the resulting $\beta$ values are divided by the initial foraging effort ($\alpha$) values such that the product of $\alpha$ and $\beta$ results in values normalized according to the degree of a species:

$$\beta_{ij}^P \Rightarrow \frac{\beta_{ij}^P}{(K_i^P)^\eta \alpha_{ij}^{\text{init}}},$$

$$\beta_{ij}^A \Rightarrow \frac{\beta_{ij}^A}{(K_j^A)^\eta \alpha_{ij}^{\text{init}}}.$$

The strength at which the network topology matters for the mutualistic interaction strength is controlled by the exponent $\eta$. A higher $\eta$ rewards specialization over generalization, and vice versa [26]. We use a value of 0.5 for all our simulations.

The resource availability is modeled based on the amount of plant resource (food) available and the demand for this resource. It is directly coupled to plant abundance and is not a state

variable on its own. The availability is represented by the supply-demand ratio $\phi_i \equiv \frac{\text{supply}}{\text{demand}}$. Pollinators compete over the resource capacity of each plant species. We assume that the plant species $P_i$ has a supply equal to its abundance: supply = $P_i$.

The demand is given by the amount of resources that the connected pollinator species aim to extract. As previously stated, per capita, pollinator species $j$ can extract $\alpha_{ij}\beta_{ij}$ from the resource. The total demand of pollinator $j$ over plant $i$ is this multiplied by the abundance: $\alpha_{ij}\beta_{ij}A_j$. The total demand for resources of plant species $i$ is given by the sum over all connected pollinators: demand $= \sum_{j=1}^{S^A} \alpha_{ij}\beta_{ij}^A A_j$.

Finally, we define a parameter $q$—called resource congestion [39]—that models the excludability of the resource. Specifically, the resource congestion, $q$, models the strength of the competition over resources by altering the demand such that

$$\phi_i \equiv \frac{\text{supply}_i}{\text{demand}_i} = \frac{P_i}{\left(\sum_{j=1}^{S^A} \alpha_{ij}\beta_{ij}^A A_j\right)^q} \,, \tag{2}$$

with $0 \leq q \leq 1$. At $q = 0$, Eq 2 equals $P_i$, meaning that there is no resource competition. For $q > 0$ there is competition over resources.

The mutualistic benefit for the pollinator species then becomes

$$\rho_j^A = \sum_{i=1}^{S^P} \alpha_{ij}\beta_{ij}^A \phi_i \,,$$

and for the plant species,

$$\rho_i^P = \sum_{j=1}^{S^A} \alpha_{ij}\beta_{ij}^P A_j \,.$$

**Migration.** Both pollinator and plant species have a positive migration rate $\mu$ to allow extinct species to regain abundance, giving rise to the full set of equations for the model (Eq 1). Migration $\mu$ is considered small so as not to influence the dynamics of the model (namely, maintaining a stable low abundance state for all species). This assumption is validated through a sensitivity analysis (see SI5).

## Adaptive foraging

The pollinators, rather than the plants, lead the choice of mutualistic interaction strengths. We assume that pollinators have a selective mechanism based on their inherent matching with plants and the resource abundance of plants. High matching and high abundance make plants more attractive to pollinators. Pollinators use adaptive foraging by changing the weights of their connections to plants to increase their gathered resources (represented in different shades in Fig 1B).

Since each pollinator seeks to maximize its gathered resources, we assume a local optimization function for $\alpha_{ij}$. We use a replicator equation (similar to [36]), given in its general form by

$$\frac{d\alpha_{ij}}{dt} = \alpha_{ij}\left(f_{ij} - \langle f \rangle_j\right) = \alpha_{ij}\left(f_{ij} - \sum_{i'=1}^{S^P} \alpha_{i'j} f_{i'j}\right),$$

with $f_{ij}$ denoting the fitness of the interaction between species $i$ and $j$. The fitness here is defined as $f_{ij} \equiv \beta_{ij}^A \phi_i$. The strength of the adaptation is controlled by a parameter $v$ ($0 \leq v \leq 1$).

This parameter generates a trade-off between keeping the original weights in the $\alpha$ matrix ($v = 1$) and purely optimizing the foraging effort for the pollinators ($v = 0$),

$$\frac{d\alpha_{ij}}{dt} = \alpha_{ij}\left(\beta_{ij}^A\phi_i - \sum_{i'=1}^{S^P}\alpha_{i'j}\beta_{i'j}^A\phi_{i'}\right)(1-v) + \left(\frac{1}{S_j^P} - \alpha_{ij}\right)v\,, \tag{3}$$

where $S_j^P$ is the number of plant species to which pollinator species $j$ is connected. Adaptive foraging decreases with increasing $v$, and there is no adaptive foraging at $v = 1$. The model is the same as described by [26] when setting $q = 0$ and $v = 1$ (i.e., no adaptive foraging and no resource competition).

The use of the replicator function is convenient since it guarantees the conservation of the total available foraging effort per pollinator species, as Eq 3 conserves the quantity $\sum_i^{S^P} \alpha_{ij} = 1$ if initialized as such. In addition, the use of replicator dynamics can also be seen as a selection process that occurs within each species for individuals with different plant preferences.

The model just described sets the foundation for a series of simulations. Next, in the Methods section, we delve into the specifics of how we operationalized these, detailing the parameters and scenarios under which the simulations were conducted.

## Methods

The baseline parameters and ranges for the simulations are given in Table A in S1 Text. The model was solved using the LSODA integrator of SCIPY with RTOL = $10^{-4}$ and ATOL = $10^{-7}$ [40].

### Feasibility of networks

Different interactions among species generate different communities. To ensure that each mutualistic network had similar characteristics, we only investigated networks that had what we call a "feasible solution", meaning that all their species survive in equilibrium when there is no external stress ($d_A = 0$) [26]. To this end, parameters were sampled from Table A in S1 Text, and the model ran to equilibrium. If all species were alive at equilibrium, the parameters were accepted. If not, the parameters were resampled up to 100 times. If, after 100 tries, no feasible parameters were found, a new network was generated. We considered species extinct when their abundance is below 0.01 [26]. The fraction of feasible networks can be seen as a measure of the persistence of the diversity of a community.

### Bifurcation-induced tipping

Bifurcation-induced tipping points were found by calculating the equilibrium trajectories of species abundance depending on the direction of change of the driver of decline $d_A$. The abundances of all species were set to initialize at 1 and evolved until equilibrium was reached for $d_A = 0$. The driver of decline $d_A$ was increased in steps of 0.02, each time running the simulation until it reached equilibrium. This continued until all pollinator species were extinct. The value of $d_A$ when the last species went extinct was denoted by the point of collapse $d_A^{\text{collapse}}$. From this extinct state, the simulation was repeated, but now the driver of decline $d_A$ was decreased each time in steps of 0.02 until reaching the point of recovery $d_A^{\text{recovery}}$: the value of the driver of decline $d_A$ at which the first species returned to existence.

The region of bistability is the region between $d_A^{\text{recovery}}$ and $d_A^{\text{collapse}}$ for each network, given $d_A^{\text{recovery}} < d_A^{\text{collapse}}$. The inequality condition guarantees that there are two stable equilibria for a given parameter set and network.

To assess the influence of the competition over resources between pollinators on the stable states of the model, the point of final collapse $d_A^{\text{collapse}}$ and first recovery $d_A^{\text{recovery}}$ of the pollinator species were calculated for different strengths of resource congestion $q$. For each resource congestion $q$, the results were averaged across 100 networks. These networks were generated as described above, except that the networks were not resampled if after 100 parameter samples the network was still not feasible, meaning even the non-feasible ones could be accepted. The reason for this is that for some values of the resource congestion not many feasible networks might exist. This also allowed us to use the fraction of feasible networks as an indicator of the persistence of the pollinator communities under a given resource congestion $q$.

### Rate-induced tipping

To identify rate-induced tipping, we checked if all pollinators went extinct at a certain critical rate of change of the driver of decline before bifurcation-induced tipping occurred. Since bifurcation-induced tipping occurred at the point of collapse $d_A^{\text{collapse}}$, the simulation was run for different rates of change $\lambda$ until a maximum value of $d_A$, denoted by the fraction $\theta$ of the point of collapse $d_A^{\text{collapse}}$ ($0 \leq \theta \leq 1$), was reached. By varying $\theta$, we could check whether the tipping response was purely due to the rate of change $\lambda$ (in which case we should see rate-induced tipping for most values of $d_A^{\text{max}}$) or due to a combination of the rate of change and bifurcation-induced tipping (in which case we should only see rate-induced tipping for a $d_A^{\text{max}}$ close to $d_A^{\text{collapse}}$). Each simulation was averaged across 100 feasible networks. The number of pollinators alive at the end of the simulation was expressed in terms of the relative persistence of the pollinator species. This is the fraction of pollinator species alive for a given rate of change $\lambda$, compared to the number of species alive at the lowest rate of change of $\lambda$ investigated. Therefore, it measures the change in the number of species alive due to increasing rates of change $\lambda$.

To test the influence of the initial abundance on rate-induced tipping, the value of the driver of decline $d_A$ at which all pollinators went extinct was calculated for different rates of change $\lambda$ of the driver of decline. To avoid confusion with the bifurcation-induced tipping point $d_A^{\text{collapse}}$, we call this point the point of extinction $d_A^{\text{extinct}}$. However, both indicate the value of the driver of decline $d_A$ that all pollinator species go extinct. Simulations were carried out for two different initial conditions: low abundance for all species ($S^{\text{init}} = 0.1$) and high abundance for all species ($S^{\text{init}} = 2$). The low abundance condition was chosen such that the abundance was in the bistable range of most pollinator species, while the high abundance condition was chosen such that most pollinator species were around or above the upper stable state (the abundant state). Each simulation was started from $d_A = 0$ with a rate $\lambda$ until all species were extinct at the point of collapse $d_A^{\text{collapse}}$, averaged across 100 feasible networks.

We used these methods to test the predictions of the model under various conditions. By systematically altering key parameters and observing the results, we aim to unravel the complex dynamics of pollinator communities in response to environmental stressors. The following Results section presents the outcomes of these simulations, shedding light on the practical implications of our model.

## Results

### Perturbed pollinator communities can exhibit rate-induced tipping with and without adaptation

We tested the behavior of the pollinator communities in response to external environmental change, considering both adaptive and nonadaptive populations with different resource congestion. If the environment is changing very slowly without adaptation, after a critical value of

the driver of decline, $d_A^{\text{collapse}}$, the pollinator community collapses (see Fig 1C). For perturbed (low initial abundance) communities without adaptive foraging, tipping occurs at specific rates, i.e., rate-induced tipping, when the driver is close to, but below, the point of collapse $d_A^{\text{collapse}}$ when initial abundances are low (Fig 2). Thus, this tipping point depends on both the magnitude and the rate of change of the driver of decline. The collapse of pollinator communities happens only around a maximum value of the point of collapse $d_A^{\text{collapse}}$ of around 70% or more (Fig 2B). In the supplementary material, Fig F in S1 Text shows the full bimodal distribution of pollinator persistence over the networks for non-adaptive communities, different rates, and resource competition.

At low rates of change, λ, the species persistence is mainly defined by the maximum driver of decline $d_A^{\max}$. Since the rate of change is slow, the system is close to equilibrium most of the time. For low rates of change and a $d_A^{\max}$ below the point of collapse $d_A^{\text{collapse}}$, this means no tipping occurs. At higher rates of change λ, the species persistence depends more on λ. In this case, all species go extinct if and only if the maximum value of the driver of decline is close enough to $d_A^{\text{collapse}}$. This is the case for almost all networks for a $d_A^{\max}$ of 80% of the point of collapse $d_A^{\text{collapse}}$.

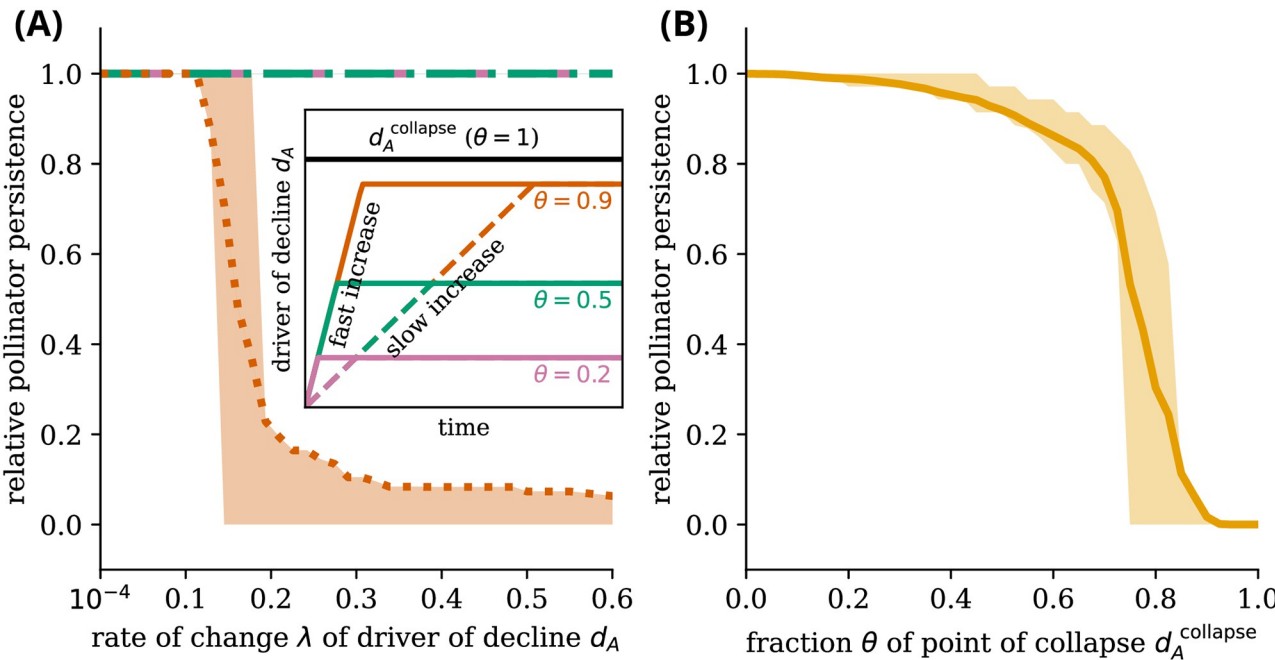

**Fig 2. Species persistence collapses for high rates of environmental change within environmental ranges that are otherwise presumed safe. (A)** Three scenarios are represented for communities without adaptive foraging, perturbed to start with a low initial species abundance (out of equilibrium). (Inset) Stress in communities increases over time as the driver of decline increases at different rates, λ, up to a maximum value, $\theta \cdot d_A^{\text{collapse}}$. The black line represents the point of collapse above which a fixed value of the driver of decline leads to the collapse of all communities and below which some communities are sustained. The maximum value of the driver of decline in each simulation is denoted by the fraction θ of the point of collapse $d_A^{\text{collapse}}$. **(A)** Dotted orange line represents an increase in the driver of decline up to 90% of the critical value, $d_A^{\text{collapse}}$, dot-dashed green line an increase up to 50%, and dashed pink an increase up to 20%. In this panel, species persistence is calculated as the fraction of pollinator species alive relative to the number of species alive at the lowest rate of change measured ($\lambda_{\min} = 10^{-4}$). **(B)** The persistence of species decreases as a function of the maximum value of the driver of decline, represented as a fraction θ of the point of collapse, for a fast rate of change (λ = 1). Communities without adaptive foraging see a critical transition in species persistence when the driver of decline increases to a value close to, but lower than, the point of collapse $d_A^{\text{collapse}}$ at a fast enough rate. In this panel, species persistence is calculated as the fraction of pollinator species alive relative to the number of species alive at θ = 0 (no external stressor). Initial species abundance $S^{\text{init}} = 0.1$ for all simulations. The results are averaged over 100 feasible networks, for which all 15 plant and 35 pollinator species survive under no stress, with the bands representing the first to third quartile ranges. Other parameters in Table A in S1 Text.

Adding adaptive foraging to the model changes the response to the rate of change, but rate-induced tipping is still present (Fig 3). For low magnitudes of the maximum driver of decline, adaptive communities already experience partial, and sometimes even full, extinction. There is a decrease in relative pollinator persistence over a large region of the fraction $\theta$ of the point of collapse $d_A^{\text{collapse}}$ (Fig 3B). In comparison, in the nonadaptive case (Fig 2B), there is a more clear threshold around $\theta \approx 0.7$ where pollinator communities go from relatively high persistence to collapse. Adaptive pollinator communities are, thus, more likely to collapse partially over a range of maximum decline drivers rather than exhibit a complete collapse of all species at a specific value of the driver (see Fig G in S1 Text for the complete bimodal distributions of pollinator persistence over the networks showing partial collapse). This suggests that adaptive pollinator communities might show a more gradual tipping process in which a few species go extinct at a time rather than a single collapse. Later, we show which species are affected throughout the gradual tipping process.

As we have shown in Figs 2 and 3, the rate of change $\lambda$ in the driver of decline $d_A$ influences the values of the driver of decline $d_A$ at which the last pollinator goes extinct. We refer to the critical value at which this rate-induced tipping occurs by the point of extinction $d_A^{\text{extinct}}$—the value of the driver of decline $d_A$ at which the last pollinator goes extinct. The point of extinction $d_A^{\text{extinct}}$ increases mostly monotonically with the rate of change $\lambda$ for high initial abundances (Fig 4). This is because, for high initial abundances, the species start with an abundance around or above the stable state with high abundance. This means species abundances change slower

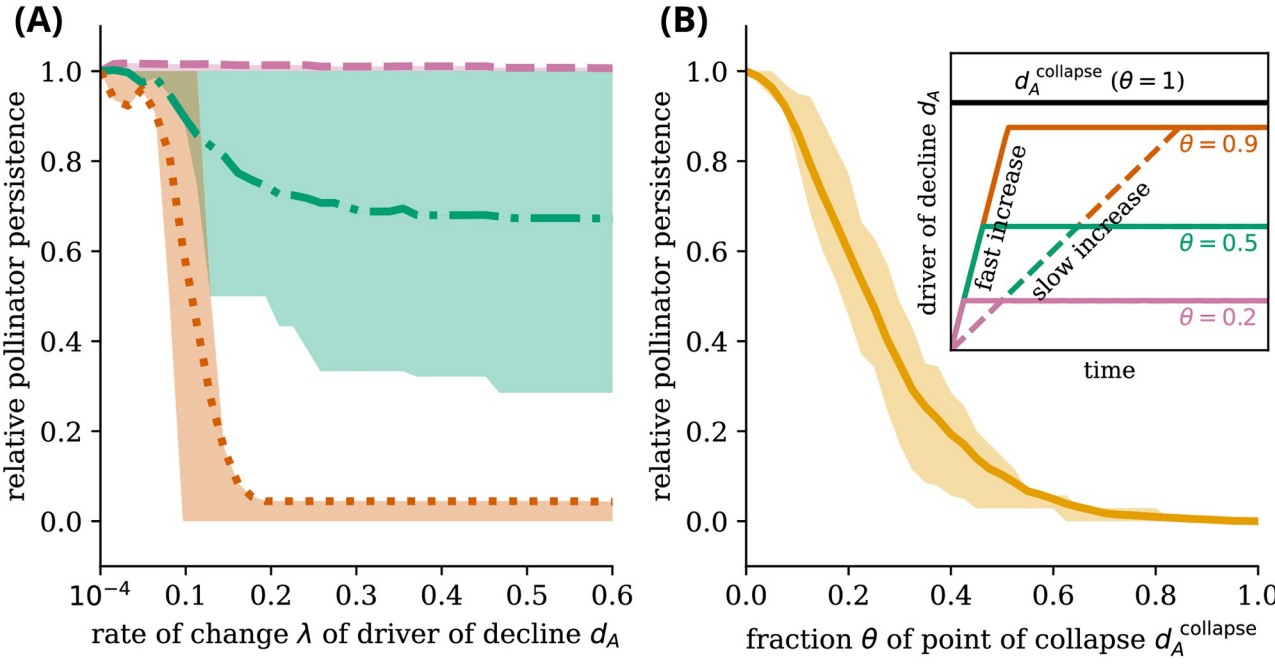

**Fig 3. Pollinator communities with adaptive foraging still collapse at high rates of change but less abruptly in the extent of environmental change.** Figure equivalent to Fig 2, but considering adaptation and resource congestion. Adaptive communities respond to an increasing driver of decline by reweighing their connections. (A) Rate-induced transitions are still present, with some communities exhibiting rate-dependent tipping at 50% of the point of collapse. Non-monotonicity is within the error range, thus, non-significant for the number of simulations. (B) Overall, pollinator persistence is more sensitive to rates of changes in a larger domain of changes in the driver of decline, $\theta$, than for communities without adaptive foraging. Some particular networks see an increase in persistence, especially for small changes and low rates of change, leading to distinct relative persistence levels above 1. For all simulations, initial species abundance of $S^{\text{init}} = 0.1$. Adaptation strength of $\nu = 0.7$ and resource congestion $q = 0.2$. The results are averaged over 100 feasible networks, for which all 15 plant and 35 pollinator species survive under no stress, with the bands representing the first to third quartile ranges. Other parameters in Table A in S1 Text.

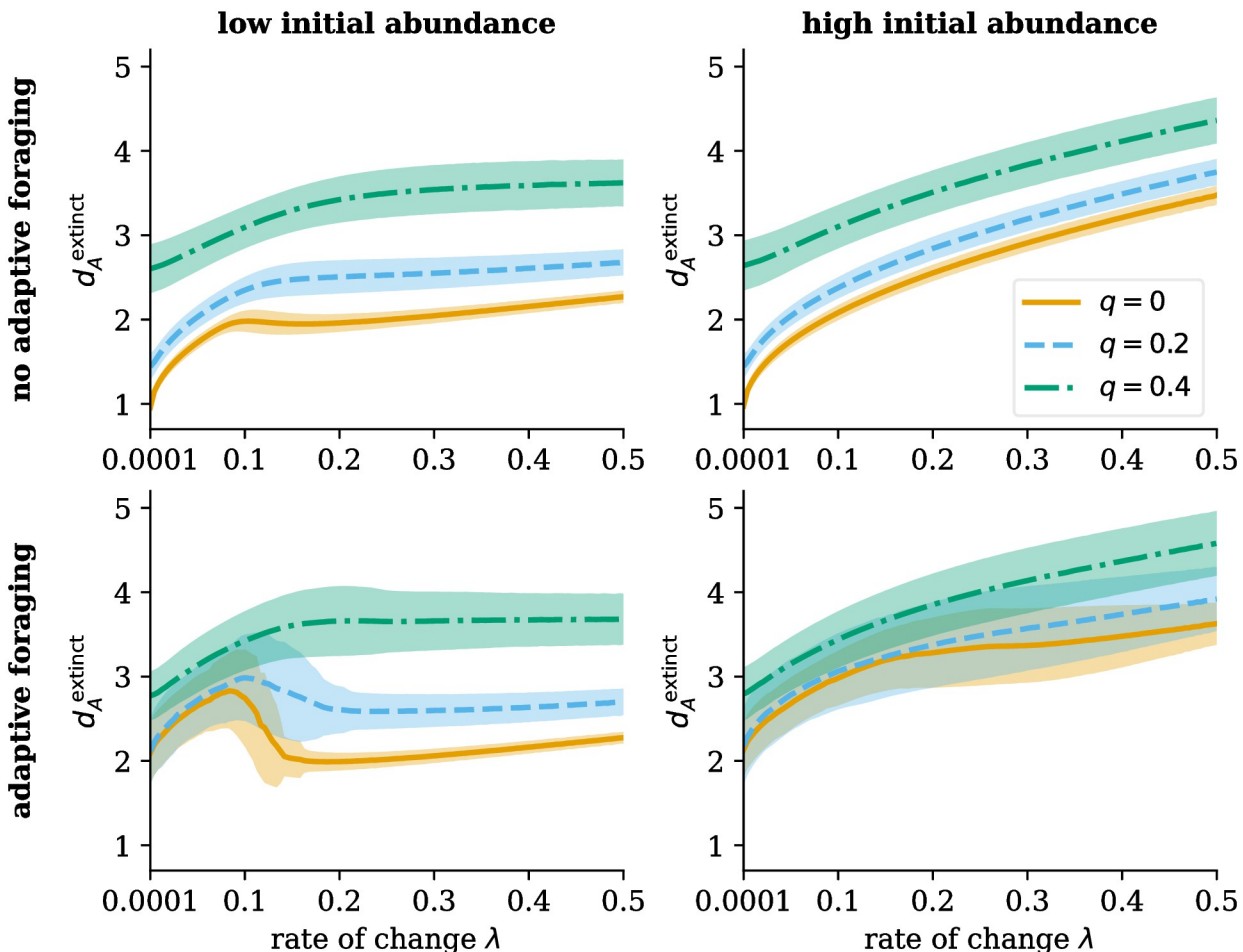

**Fig 4. Higher rates of change in the driver of decline lead to extinction at lower values of the driver of decline only for perturbed communities.** The figure shows the value of the driver of decline $d_A$ at which all pollinators go extinct, $d_A^{\text{extinct}}$, as a function of the rate of change $\lambda$ of the driver of decline. For a low initial pollinator abundance (left panels), after a critical value of the rate of change $\lambda$, $d_A^{\text{extinct}}$ has a nonlinear response. This effect disappears for higher resource congestion $q$, while it increases with stronger adaptation. For a high initial pollinator abundance, $d_A^{\text{extinct}}$ increases monotonically with the rate of change $\lambda$. The results are averaged over 100 feasible networks per case (lines), shown with the standard deviation across networks (band). $v = 0.7$ for the case with adaptive foraging. The initial abundance for all species is $S^{\text{init}} = 0.1$ for the low initial abundance condition and $S^{\text{init}} = 2$ for the high initial abundance condition. Other parameters in Table A in S1 Text.

than the driver of decline $d_A$. In other words, the system responds slower to forcing than the forcing itself changes. In this case, rate-induced tipping is not present.

On the other hand, at low initial abundance, there is a non-monotonic response of the point of extinction $d_A^{\text{extinct}}$ as a function of the rate of change $\lambda$. More specifically, inside the bistable region of the system (between the two stable states)—where species have a low initial abundance—higher rates of change $\lambda$ can lead to a lower $d_A^{\text{extinct}}$ after a critical value of $\lambda$ as shown in Fig 4. This supports the notion that the pollinator communities need to be perturbed for rate-induced tipping to be possible. Furthermore, Fig 4 shows a stronger non-monotonic behavior for pollinator communities with adaptive foraging than without adaptive foraging.

The collapse of pollinator communities can thus not be fully understood by bifurcation-induced tipping or rate-induced tipping separately. First, the system must possess bistability for rate-induced tipping to occur. Without bistability, there is no opportunity to tip from one

branch to another since there is only one branch. High rates of change can then, in this case, not force the system to tip from one state to another. Second, the presence of rate-induced tipping depends on the trajectory in state-space. This "tipping trajectory" depends on i) the rate of change of the driver of decline, which needs to be high enough (Figs 2A and 3A), ii) the total magnitude of the driver of decline, which needs to be close enough to the point of collapse as dictated by the equilibrium trajectories (Figs 2B and 3B), and iii) the perturbation applied to the system (e.g., the initial abundance of species), which needs to be below the upper stable branch in the bistable area of the bifurcation diagram (Fig 4).

## Influence of adaptation and congestion on stability

To better illustrate how adaptive foraging alters the dynamics under increasing stress, Fig 5 shows a representative simulation of plant and pollinator abundances under increasing drivers of decline $d_A$. The plant and pollinator species start with a low abundance ($S^{init} = 0.1$). The driver of decline $d_A$ is increased from 0 with a rate of $\lambda = 0.05$. Given the slow increase in the driver of decline $d_A$, both plant and pollinator species initially recover their abundances (Fig 5A and 5B). Most plant and pollinator species reach their peak around 10 time units after which their abundances decrease. Eventually, all pollinator species go extinct.

During this process, pollinators shift their foraging effort so that it benefits them the most. Panels C-H of Fig 5 show how their foraging effort changes, sorted by their number of connections (degree). After roughly 10 time units, the driver of decline increases to such a level that many species start to deteriorate. At this point, some large changes occur in the foraging effort. Most notably, a few pollinators manage to invest heavily in one or two plant species, thereby temporarily offsetting their decline in abundance. This leads some plant species to stabilize their decline for an extended period, which allows certain high-degree pollinator species to persist significantly longer than other, lower-degree pollinator species. The two pollinator species with degree 9 (Fig 5H) even temporarily benefit from the extinction of other species, as can be observed by a temporary increase in their abundances (Fig 5A). They increase their foraging effort significantly in one of their connected plant species, leading to a loss of biodiversity in plants while simultaneously promoting two plant species (Fig 5B). At the same time, many low- to mid-degree pollinator species do not manage to adapt their foraging effort in such a way as to offset the negative effect of the driver of decline. Eventually, the pressure of the driver of decline gets so high that all pollinator species go extinct.

This representative simulation also gives an explanation for the curve in Fig 3B where at low $\theta$ values the relative pollinator persistence already decreases before reaching 0 between $0.4 < \theta < 0.6$. As Fig 5A illustrates, some pollinator species go extinct faster than others (faster relates to a lower driver of decline value). In combination with Fig 3, where the simulations ran until equilibrium, it shows that partial pollinator community collapse occurs when there is adaptive foraging.

Both the adaptation strength $v$ and the resource congestion $q$ influence these dynamics and, thus, the bifurcation diagram of the pollinator abundances. Three examples are shown in Fig 6. Without adaptation, both pollinator species collapse and recovery are abrupt as a function of the control parameter driver of decline $d_A$. Adding resource congestion leads to an almost complete disappearance of bistability in this area due to the disintegration of the network structure. Adding adaptation leads to the reintroduction of bistability. At the same time, both the collapse and the recovery of pollinators become less abrupt. Looking at individual species, as illustrated in S1 Text Figs B-E, some high-degree and highly abundant species can survive longer after the partial collapse of the system. We see a similar effect in the adaptive pollinator community in Fig 5. Some highly abundant, generalist species get a temporary boost after the

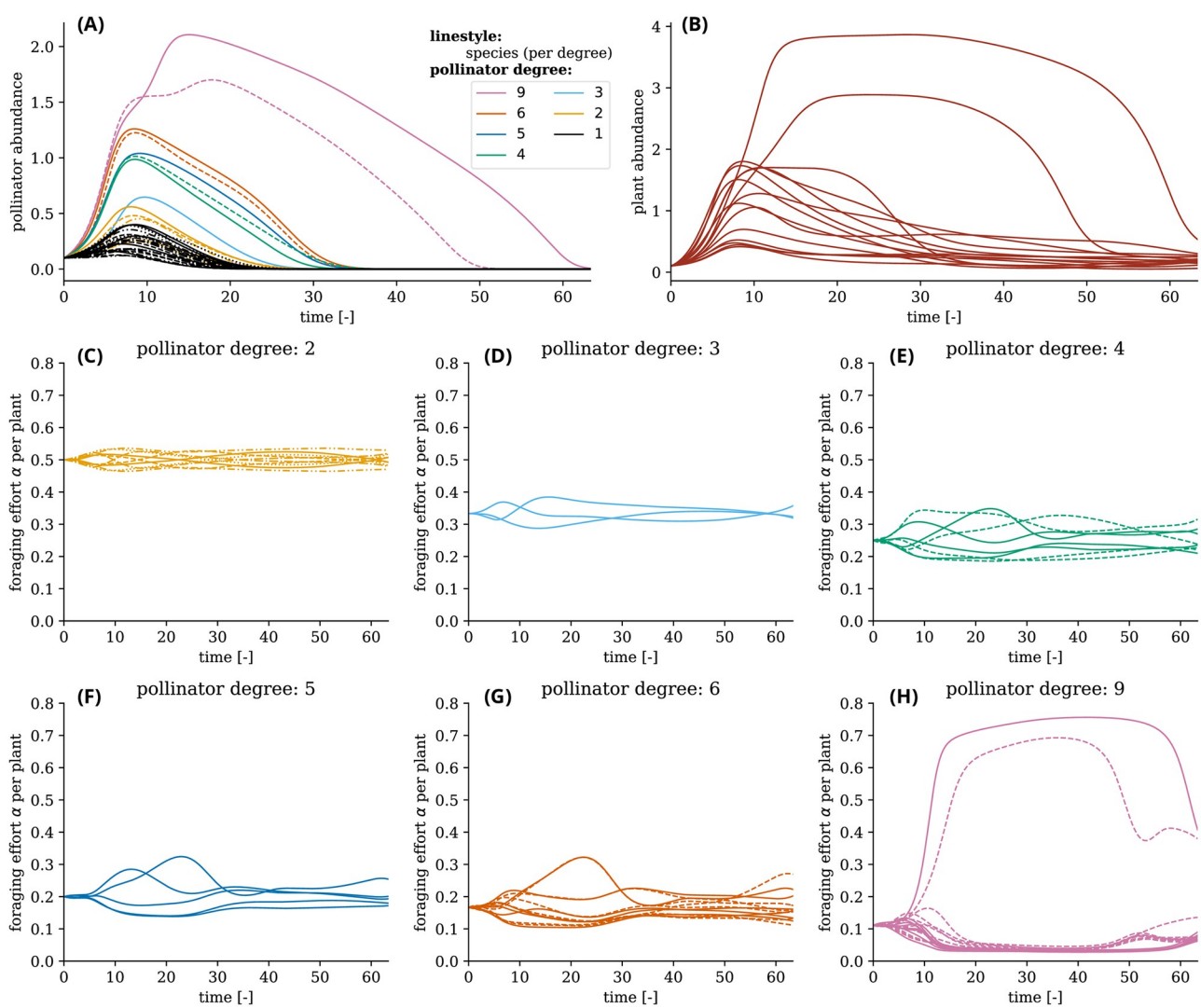

**Fig 5. The coevolution of foraging effort and species abundances under increasing driver of decline $d_A$.** The top row shows the evolution of pollinator and plant abundances under linearly increasing driver of decline $d_A$ with rate $\lambda = 0.05$, starting from a low abundance condition of $S^{\text{init}} = 0.1$, with adaptation strength of $\nu = 0.7$ and resource congestion $q = 0.2$. Each line style and color combination represents a single pollinator species in all graphs (except the plant abundance graph, top right). For example, there is one pollinator species with degree 3. Since the degree is 3, there are three solid blue lines (one for each connection to a plant species). Another example is the two pollinator species with degree 4, thus showing eight lines (four solid lines for one species and four dashed lines for another species). Since the values of the foraging effort $\alpha$ for each individual species add up to 1, the evolution of the foraging effort $\alpha$ is not shown for pollinators with degree one since they have a constant $\alpha = 1$ to their single connected plant species. Pollinators with high degree rapidly become the most abundant. Furthermore, the foraging effort $\alpha$ drastically changes—especially around 10 time units when most species reach their peak abundance. The two pollinator species with degree 9 survive the longest and also have one plant species in which they invest most of their foraging effort after 10 time units. Other parameters in Table A in S1 Text.

partial collapse. This process leads to a more gradual decrease in the average pollinator abundance, as shown in Fig 6.

The resource congestion $q$ (dictating how strong the competition over resources is) determines whether pollinator communities possess hysteresis. As Fig 7 shows, hysteresis is always present in pollinator communities without resource competition. However, increasing resource congestion leads to a loss of the bistable states that define hysteresis in both the model with and without adaptive foraging, hereby merging the point of collapse $d_A^{\text{collapse}}$ and the point

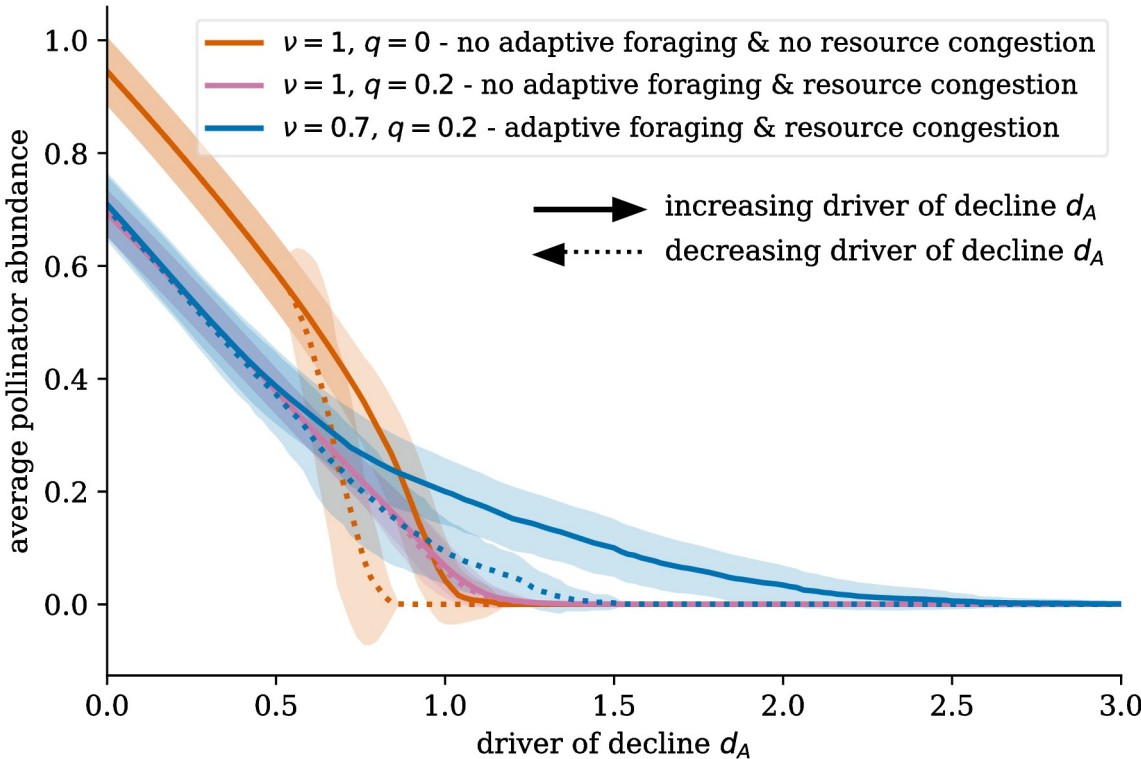

**Fig 6. Influence of adaptive foraging and resource congestion on the bifurcation diagram.** Adaptation changes the bifurcation diagram. The collapse is less abrupt for pollinator communities with adaptive foraging. Resource competition decreases the size of the bistable area if there is no adaptive foraging. S1 Text Figs B-E show the full bifurcation diagrams of all individual pollinator species for different settings of $v$ and $q$. The results are averaged over 100 networks per parameter setting, with the error bands showing the standard deviation. Parameters in Table A in S1 Text.

of recovery $d_A^{recovery}$. Bistable states disappear with higher resource congestion $q$ for systems with adaptive foraging compared to systems without adaptive foraging.

The increase of the driver of decline of collapse $d_A^{collapse}$ and recovery $d_A^{recovery}$ is illustrated by the bifurcation diagrams shown in S1 Text Figs B-E. With increasing resource congestion $q$, some pollinator species have a lower abundance at zero stressors and go extinct faster. In contrast, other species have a slower and more gradual decrease in abundance once the resource congestion $q$ increases. Due to the increased competition for resources, the stress on some species is high, which causes them to go extinct fast. However, the longest-surviving species have little competition over their resources with a high driver of decline $d_A$, making them less prone to extinction. This is because when there is low competition over resources, there is a high supply-demand ratio, which boosts the growth rate of these species. On the downside, this results in a significant loss of species diversity. In general, the tipping points become less synchronized as resource congestion $q$ increases.

In the case of adaptive foraging, competition over resources helps pollinators spread their foraging effort more evenly by increasing the profits for plants with few interactions. However, once competition over resources increases too much, some plant and pollinator species cannot survive. This makes secondary extinctions possible, which depend on the extinct species. This process leads to more infeasible networks.

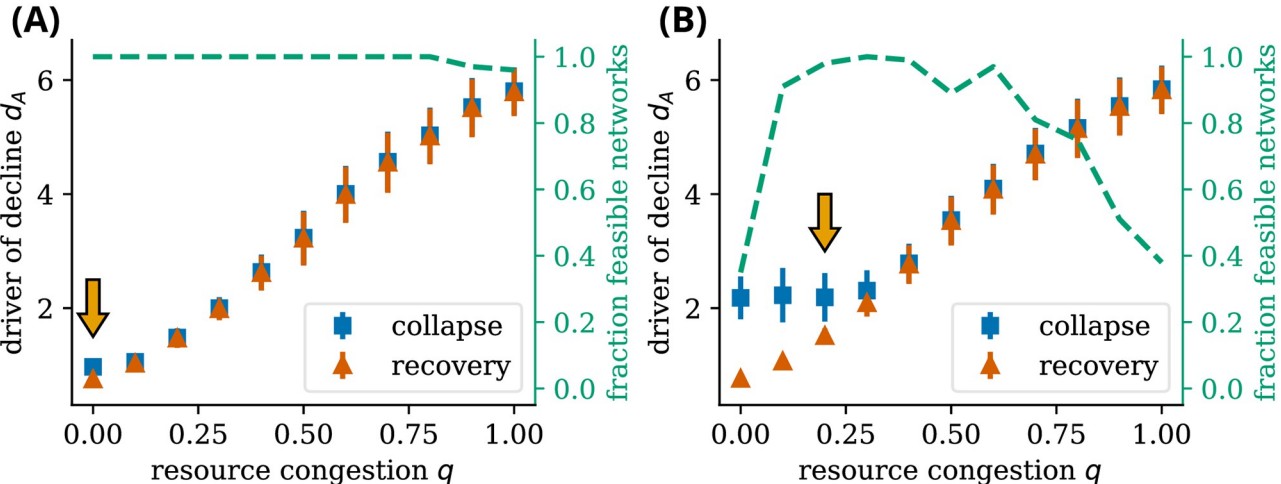

**Fig 7. Adaptability and resource congestion affect hysteretic patterns and the viability of plant-pollinator networks. For adaptive pollinators, intermediate levels of resource congestion increase the overall persistence of ecological networks**. The point of collapse and recovery of pollinator species increases as a function of resource congestion $q$ without (A) and with adaptive foraging (B). For low resource congestion, the system possesses bistable states which disappear after a critical value of the resource congestion. Resource congestion also affects the feasibility of the networks—networks for which all 15 plant and 35 pollinator species survive under no stress. An intermediate level of resource congestion is required for the adaptive model to produce feasible networks. The orange arrows indicate the resource congestion strength $q$ chosen for the simulation of Figs 2 and 3. These values were chosen such that the systems possess bistable states—as observed in the non-overlapping points of collapse and recovery—and have a high fraction of feasibility. For low resource congestion, adaptability increases the range of drivers of decline at which pollinator communities do not collapse, increasing resilience in the Holling sense [41]. (A) $v = 1$ and (B) $v = 0.7$. The results are averaged over 100 networks per value of resource congestion $q$ with the error bars showing the standard deviation. Other parameters in Table A in S1 Text.

## Influence of adaptation strength

The feasibility of the networks also depends greatly on the strength of adaptation $v$. At higher adaptation, fewer species survive because a few plant species gather high foraging effort from pollinators, boosting these plants' abundance. Thus, more pollinators pollinate these highly abundant plant species and lose interest in other plant species, leading to their extinction. This, in turn, causes the extinction of pollinator species that depend on these now-extinct plants. When there is a high level of adaptation and only a few generalists with high abundances, only the pollinators connected to them survive. Although this could lead to a strong increase in abundance in these selected plant and pollinator species, the structure of the network is also highly affected. Due to the extinction of many specialist plants, the network structure disintegrates. This leads to the disappearance of highly synchronized extinction events (this is apparent in Fig D in S1 Text: at high adaptation, there are many extinction events of just a few pollinators per time, as opposed to the more synchronous tipping in Fig A in S1 Text). Due to both the higher individual abundances of successful species and less synchronized tipping, a higher driver of decline $d_A$ is necessary to kill all pollinator species. At the same time, pollinators not connected to these generalists either do not survive under zero external stressors—leading to infeasible networks—or already go extinct at low values of the driver of decline $d_A$.

A Sobol sensitivity analysis in SI5 shows the model parameters' influence on the networks' feasibility and the point of collapse. The resource congestion $q$ and adaptation strength $v$ have the strongest influence in both cases. Furthermore, the migration rate $\mu$ influences the results very little, thereby validating our assumption that this parameter does not significantly influence the dynamics of the model.

The above results provide a detailed account of how pollinator communities react under different stress scenarios, highlighting the critical role of adaptation and the rate of environmental changes. In the Discussion, we interpret these findings in a broader context, discussing their implications for ecological theory, pollinator conservation strategies, and future research directions.

## Discussion

The species that form pollinator communities are codependent and interact in complex ways. This makes it challenging to study how the rate of change of external environmental forces affects the stability of a community. Some of the existing models do not capture adaptive processes and, therefore, are inadequate to understand how changing environments affect pollinator communities. Here, we provide a model that includes adaptation and, thereby, the capacity to respond to change, which can be used to understand the future state of pollinator communities more accurately.

With this model, we shed light on overlooked perils, such as rate-induced tipping. We demonstrate that rate-induced tipping is present in a theoretical model excluding and including adaptive foraging in pollinator communities. Rate-induced tipping points, where all pollinators go extinct, can occur for values of the driver of decline whose effects are otherwise considered reversible when the driver of decline is constant or slowly varying. For rate-induced tipping to occur, the system needs to possess bistable states and be perturbed out of equilibrium.

The existence of bistability in pollinator communities is strongly influenced by both competition over resources and adaptive foraging, highlighting the importance of including natural mechanisms in models to anticipate the responses of real ecosystems. Here, we show that high resource competition—through high resource congestion—can lead to the disappearance of hysteresis. In other words, there is only one stable state for each value of the driver of decline, whereby tipping is no longer present. On the other hand, in the case of adaptive foraging, resource competition is vital to create feasible networks. Without resource competition, only a small fraction of networks are feasible in adaptive pollinator communities.

Importantly, the adaptive capacity of pollinator communities influences rate-induced tipping. Systems possessing adaptive foraging have a stronger response to different rates of change of the driver of decline such that some pollinator species already go extinct at slow rates. This is because rapid adaptation can temporarily mask the negative effects of an increasing driver of decline, but it can also lead to additional competition for resources, which can negatively affect some species by reducing their abundance.

The evidence for community-scale collapse draws mainly from theoretical work, without direct empirical observations of such an event [23]. This lack of empirical observation could be due to the rarity of such an event. Indeed, our work supports the idea that community-scale collapse—in its traditional narrow sense—is rare because species in adaptive pollinator communities are unlikely to disappear simultaneously. We were able to gain this insight with our model for two reasons. First, as opposed to previous studies that looked at static external environments, the two timescales (that of the external environment and that of the pollinators) are coupled. Second, by including adaptive foraging, we include part of the natural adaptive capacity of pollinator communities to react to environmental change.

Notably, here we speak of rate-induced tipping and bistability as community concepts, showing how the critical values of the rate of change of driver of decline can differ between pollinator communities. However, when extended to the species scale, this region is different for each species, with different species collapsing under different conditions. The different

species are heterogeneous, have asymmetric interactions, and, consequently, do not all react equally to environmental degradation. What we observe in our model with adaptive foraging is a sequence of "failures"—tipping events of single or small groups of pollinator species—before the whole community disappears. Some species abundances—mainly generalist species—even see a temporary boost after the extinction of a few species due to the decrease in competition over resources. Therefore, generalists are more robust to rapid environmental changes than specialists. The fact that specialist species are the first to disappear seems to align with empirical studies [10, 23, 42].

The new model more accurately represents a variety of mechanisms present in real pollinator communities, such as added competition over resources through the supply-demand ratio and adaptive foraging. Our study can capture adaptation on ecological timescales by including adaptive foraging through resource competition and explicit environmental change. Nevertheless, there are other adaptive mechanisms in pollinator communities besides adaptive foraging that we did not investigate. Two of them are (co-)evolution and the rewiring of species.

Evidence shows that co-evolution occurs between plant and pollinator species [43]. Plants can adapt through evolution by genetic variation [44], being able to do this quickly when pollinator communities change suddenly. Co-evolution leads to different networks [45]. In ecosystem models in general, adaptive responses at the genetic level can play an important role in evaluating the response to changing external stressors [46]. Evolution may be vital on larger timescales when the environment changes slowly enough for evolutionary mechanisms to activate or when evolution is fast [47–49].

Moreover, contrary to our assumption of a fixed set of possible interactions between plant and pollinator species, these species can also form novel interactions through rewiring by forming new connections with existing or immigrant species [32]. This contrasts with our definition of adaptive foraging, where only the strength of interactions changes, but no new connections are possible. Rewiring occurs especially from pollinators to plants that have high abundances and when the trait matching is increased between them [34]. Rewiring of the species can further increase resilience [50]. It can also help pollinator communities adapt to strong phenological changes [10]. Furthermore, species are more resistant to extinction due to the rewiring of species interactions in mutualistic networks [51]. Yet, the effect of rewiring on rate-induced tipping is still unknown. Without this knowledge, we might have underestimated the resilience of pollinator communities and, at the same time, found a smaller role for adaptation in rate-induced tipping. Therefore, the conclusions of this study stem from a pessimistic scenario but are nevertheless important for calibrating preventive measures so that pollinator communities remain far from extinction.

Including all these effects to add realism comes with the drawback that models with higher effective dimensions tend to produce more uncertain estimates, especially when data are scarce [52]. Thus, a parallel effort of mechanism calibration is necessary.

We show that including sufficient natural mechanisms in models describing pollinator communities changes the resilience of these systems, opening perspectives for the management of real ecosystems. Introducing resource competition leads to two different regimes: one in which hysteresis is present and one in which it is not. Although the highest rate of feasible ecological networks is found in systems with hysteresis, based on our results, it cannot be ruled out that there are pollinator communities without hysteresis. This should be considered when estimating the effort needed to restore pollinator communities from collapse. Restoring systems without hysteresis is easier than restoring systems with hysteresis since the former is reversible while the latter is not.

So far, ecosystem management has been concerned with establishing limits to species abundance and limits to the external environmental conditions of these ecosystems. For instance,

the group on the post-2020 Global Biodiversity Framework mentions the goal of a "reduction of ecological footprint in an equitable manner (. . .) within planetary boundaries by 2050" [53]; other works advocate the use of fixed targets as opposed to net rates [54]. The work on rate-induced transitions, to which our study contributes, shows that to determine targets of safe operation, it is important to consider not only the extent but also the rate of change in external conditions. Only then can we make a correct assessment of the resilience and recovery of these ecosystems. By neglecting that external conditions change over time, alterations in dynamics due to the rate of change are overlooked [19]. In practice, the rate of change can be a boundary in itself.

Since we showed that perturbed pollinator communities can experience rate-induced tipping, rate-induced tipping might be expected under stochastic forcing [27]. The next question to be explored is what the risk is for rate-induced tipping under stochastic forcing.

To further test the disappearance of the bistability due to resource competition, it would be interesting to examine existing models that include resource dynamics, e.g., [36] and [55], for the presence of hysteresis as a function of induced mortality through added drivers of decline. These models, with higher complexity, have explicit resource dynamics and some type of adaptive foraging, instead of the implicit parameterized resource dynamics used in our study. The conditions for the presence or absence of hysteresis in these models with more fundamental processes would make it clearer when pollinator communities possess bistable states and whether they experience rate-induced tipping.

Beyond further extensions, further analysis of this model could also provide additional insights into the resilience of pollinator communities. In our model, some specific networks saw an increase in relative pollinator persistence with environmental change. Systematically identifying the properties of these networks could also contribute to a better understanding of what structures improve pollinator communities resilience. Additionally, the analysis of possible different extinction pathways and consequent multi-stability could also shed light on the path dependency of these communities.

Finally, our work also has implications for the study and notions of resilience. Since Holling [41], the concept has evolved from local bounce-back speed metrics to the ability of systems to reorganize under change and find new (but productive) equilibria. In the context of pollinator communities, these concepts are useful in defining resilience in terms of determining the region of external conditions for plants and pollinators that allow the system to perform such a transformation [56]. However, these external conditions cannot be seen (quasi-)statically. New uses of resilience must be sure to incorporate rapid external change.

## Supporting information

**S1 Text. Text with supporting information.** The text with supporting information contains additional computational experiments and algorithms. It is divided into five sections. **Section A. Simulation set-up**. Includes: **Table A. Default model parameters**. The default parameter values and ranges used in all simulations, unless otherwise specified. AF = Adaptive Foraging, $\sim U(\cdot)$ = drawn from a uniform distribution at the beginning of each simulation. **Section B. Network generation**. Describes the network generation algorithm and contains: **Fig A. Adjacency matrix of a nested network and forbidden links** Adjacency matrix of a pollinator network on the left with the corresponding forbidden links matrix on the right. Black squares denote the presence of a link. The connectance is 0.15 and the fraction of forbidden links is 0.3. There is a clear difference visible between generalist species and specialist species, in the sense that there are a few species with high connectivity and many with low connectivity. **Section C. Dependence of hysteresis on resource congestion and adaptation**. Contains various

additional computation experiments. **Fig B. Hysteresis for increasing resource congestion $q$ for the non-adaptive model**. Equilibrium abundance of pollinator species as a function of the drivers of decline $d_A$ for increasing resource congestion $q$ for the non-adaptive model. The blue lines show the equilibrium trajectory for increasing $d_A$ and the orange lines show the equilibrium trajectory for decreasing $d_A$. See Table A for the parameters used. **Fig C. Hysteresis for increasing resource congestion $q$ for the adaptive model with $v = 0.8$**. Equilibrium abundance of pollinator species as a function of the drivers of decline $d_A$ for increasing resource congestion $q$ for the adaptive model with $v = 0.8$. The blue lines show the equilibrium trajectory for increasing $d_A$ and the orange lines show the equilibrium trajectory for decreasing $d_A$. See Table A for the parameters used. **Fig D. Hysteresis for increasing resource congestion $q$ for the adaptive model with $v = 0.7$**. Equilibrium abundance of pollinator species as a function of the drivers of decline $d_A$ for increasing resource congestion $q$ for the adaptive model with $v = 0.7$. The blue lines show the equilibrium trajectory for increasing $d_A$ and the orange lines show the equilibrium trajectory for decreasing $d_A$. See Table A for the parameters used. **Fig E. Hysteresis for increasing resource congestion $q$ for the adaptive model with $v = 0.6$**. Equilibrium abundance of pollinator species as a function of the drivers of decline $d_A$ for increasing resource congestion $q$ for the adaptive model with $v = 0.6$. The blue lines show the equilibrium trajectory for increasing $d_A$ and the orange lines show the equilibrium trajectory for decreasing $d_A$. See Table A for the parameters used. **Section D. Distribution of pollinator persistence**. Describes the distribution of pollinator persistence in different computational experiments. **Fig F. The full distribution of relative pollinator abundance for three different rates of change λ accompanying** Fig 2A **in the paper (no adaptive foraging)**. $\theta$ is the fraction of the point of collapse $d_A$ at which point the relative pollinator persistence is measured. The distributions are bimodal around 0 and 1 which indicates that there is an abrupt collapse of networks at increasing rates of change. See Table A for the parameters used. **Fig G. The full distribution of relative pollinator abundance for three different rates of change λ accompanying** Fig 2A **in the paper (with adaptive foraging)**. $\theta$ is the fraction of the point of collapse $d_A$ at which point the relative pollinator persistence is measured. The distributions are mainly bimodal around 0 and 1. However, some networks have a persistence between 0 and 1, indicating partial collapse due to the rate of change. Furthermore, there are a few networks with pollinator persistence significantly above 1, indicating nonlinear effects where sometimes individual networks can profit from higher rates of change. See Table A for the parameters used. **Section E. Sensitivity analysis**. Contains: **Table B. Parameters for the sensitivity analysis on the feasibility of networks**. Parameters and their value ranges used for the sensitivity analysis on the feasibility of networks, and plant and pollinator abundances. The fixed parameters can be found in Table A. AF = Adaptive Foraging. **Fig H. Sensitivity analysis of the number of plant species alive**. Sobol sensitivity analysis of the number of plant species alive depending on five parameters: resource congestion $q$, nestedness $N$, connectance $D$, adaptation strength $v$, and migration rate $\mu$. The sample size per parameter was 512. The adaptation strength $v$ had the strongest effect on the variance of the outcome of the model. **Fig I. Sensitivity analysis of the number of pollinator species alive**. Sobol sensitivity analysis of the number of pollinator species alive depending on five parameters: resource congestion $q$, nestedness $N$, connectance $D$, adaptation strength $v$, and migration rate $\mu$. The sample size per parameter was 512. The adaptation strength $v$ had the strongest effect on the variance of the number of pollinators alive. The migration rate $\mu$ only has a marginal effect. **Fig J. Sensitivity analysis of the total number of species alive**. Sobol sensitivity analysis of the total number of species alive depending on five parameters: resource congestion $q$, nestedness $N$, connectance $D$, adaptation strength $v$, and migration rate $\mu$. The sample size per parameter was 512. The adaptation strength $v$ had the strongest effect on the variance of the outcome of the model. **Fig K.**

**Sensitivity analysis of the abundance of plant species**. Sobol sensitivity analysis of the average plant abundance depending on five parameters: resource congestion $q$, nestedness $N$, connectance $D$, adaptation strength $v$, and migration rate $\mu$. The sample size per parameter was 512. **Fig L. Sensitivity analysis of the abundance of pollinator species**. Sobol sensitivity analysis of the average pollinator abundance depending on five parameters: resource congestion $q$, nestedness $N$, connectance $D$, adaptation strength $v$, and migration rate $\mu$. The sample size per parameter was 512. **Table C. Parameters for the sensitivity analysis on the critical driver of decline of collapse $d_A^{\mathrm{collapse}}$**. Parameters and their value ranges used for the sensitivity analysis on the critical driver of decline of collapse $d_A^{\mathrm{collapse}}$. The fixed parameters can be found in Table A. AF = Adaptive Foraging, $d_A$ = driver of decline. **Fig M. Sensitivity analysis on the driver of decline $d_A$**. Sobol sensitivity analysis on the value of driver of decline at which all pollinator are extinct $d_A^{\mathrm{collapse}}$, depending on six parameters: resource congestion $q$, nestedness $N$, connectance $D$, adaptation strength $v$, initial abundance per species $S^{\mathrm{init}}$, and migration rate $\mu$. The sample size per parameter was 512. (PDF)

# Acknowledgments

The authors acknowledge the fruitful discussions and stimulating environment that occurred during the 'BIRS workshop [22w5067]: Rate-Induced Transitions in Networked Systems.' A special thanks to the several discussions with Guido Vaessen, who provided continuous comments throughout, and to Theresa W. Ong, Kristen M. Jovanelly, Chenyang Su, and Wenying Liao, who provided initial comments on the research. A further thank-you note to Mike Lees, who provided valuable and motivating comments on the research and initial versions of the figures and original manuscript. ST acknowledges the support of the Institute of Advanced Study (University of Amsterdam) where part of the work was carried out.

# Author Contributions

**Conceptualization:** Sjoerd Terpstra, Flávia M. D. Marquitti, Vítor V. Vasconcelos.

**Formal analysis:** Sjoerd Terpstra, Vítor V. Vasconcelos.

**Funding acquisition:** Vítor V. Vasconcelos.

**Investigation:** Sjoerd Terpstra, Vítor V. Vasconcelos.

**Methodology:** Sjoerd Terpstra, Flávia M. D. Marquitti, Vítor V. Vasconcelos.

**Project administration:** Vítor V. Vasconcelos.

**Software:** Sjoerd Terpstra.

**Supervision:** Vítor V. Vasconcelos.

**Validation:** Sjoerd Terpstra, Flávia M. D. Marquitti, Vítor V. Vasconcelos.

**Visualization:** Sjoerd Terpstra, Vítor V. Vasconcelos.

**Writing – original draft:** Sjoerd Terpstra, Vítor V. Vasconcelos.

**Writing – review & editing:** Sjoerd Terpstra, Flávia M. D. Marquitti, Vítor V. Vasconcelos.

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
