## [Decision Letter · Decision Letter 0]

15 May 2023

Dear Dr. Vasconcelos,

Thank you very much for submitting your manuscript "Early irreversible collapse of perturbed pollinator communities under rapid environmental change" for consideration at PLOS Computational Biology.

As with all papers reviewed by the journal, your manuscript was reviewed by members of the editorial board and by independent reviewers. In light of the split reviews (below this email), we cannot accept the manuscript in its current form but we would like to invite the resubmission of a significantly-revised version that takes into account the reviewers' comments, in particular those of reviewer 1 considering the writing, motivation and implications of the work.

Please consider the question of whether a rate-induced tipping point is to be expected given the existence of bi-stability, and the recommendation that the novelty of the work resides on the consideration of adaptive foraging.  Also, reviewer 1 raises the question of any empirical evidence for community collapse in these kinds of systems that would motivate the focus of the study.  If that evidence is not there, we would recommend a clear emphasis on the theory, especially on the focus on adaptive foraging, and the more clear presentation of the relevance of this kind of transition vs. the dynamical properties expected from bi-stability and an external driver. We also agree with the reviewer that the implications for conservation need to be presented in a more careful and clear way.  Both reviewers have made recommendations on the writing which will be helpful.

We cannot make any decision about publication until we have seen the revised manuscript and your response to the reviewers' comments. Your revised manuscript is also likely to be sent to reviewers for further evaluation, and we would like to see at that stage more substantive support from reviewers about publication in this journal. 

Sincerely,

Mercedes Pascual

Academic Editor

PLOS Computational Biology

Natalia Komarova

Section Editor

PLOS Computational Biology

Reviewer's Responses to Questions

**Comments to the Authors:**

Reviewer #1: This is a theoretical paper, which builds on previous studies (in particular Lever et al. 2014) investigating the emergence of community scale tipping points in a plant-pollinator model. Here, the authors add adaptive foraging of the pollinators in the model (i.e. pollinators can adapt their relative investments in their plant partners through time) and ask whether this mechanism affects the probability of community-scale tipping. The authors are in particular interested in rate-induced tipping: in the simulations, the driver of degradation (which is a linear loss term for the pollinators) can either be fixed or change in time, which allows investigating how the relative time scales of the processes affect tipping. One of the original aspects of the study is that the foraging effort is itself considered to be a dynamical variable.

I found the study interesting from a theoretical point of view. However, the justification of the study (the main objective and novelty of the paper) and the interpretation of the results are very misleading.

The way things are presented gives the impression that the study is based on strong empirical support, while it's not. This is an extension of a previous modeling study (Lever et al. 2014), which was itself not based on any empirical studies. As far as I know, there hasn’t been a single observation of a community-scale tipping point in a plant-pollinator community. As mentioned in Latty 2019 (cited in the paper): « We found only 1 empirical example of a tipping point or threshold response to stress at the colony level (Bryden et al 2013), and no empirical examples at the population or community level.” Also, it’s noteworthy that, even in models, community-scale tipping points are an exception rather than the rule. It only happens for certain network structures (and very specific dynamical equations). Therefore, it’s important to realize that the setting of the study is a very particular one. This does not mean that this work is not interesting, but this means that the justification of the study must be rewritten to justify why it’s relevant to investigate the emergence of community-scale tipping points in a plant-pollinator community model. Also - and maybe even more importantly - the references to applications of this work for conservation must be presented much more carefully.

Furthermore, the novelty of the work is not really clear in this version of the text. The way the text reads suggests that the novelty is that rate-induced tipping is found in the model. However, rate-induced tipping can happen in any dynamical system with bistability. So, there is no surprise that this can happen in this specific model too. To me, the novelty resides in the introduction of adaptive foraging in a plant-pollinator model which has been shown to exhibit tipping point and to investigate the effect of this mechanism on the likelihood of tipping. I would rewrite the introduction with a much stronger focus on the foraging mechanism (why introducing this mechanism rather than another one, what are the evidence that it is particularly relevant from plant-pollinator systems, why looking at the links between foraging and tipping, etc). I would therefore also expect to see more results about the effect of adaptive foraging. For example, how does introducing adaptive foraging affects the bifurcation diagram of the model, has the location of tipping changed when adaptive foraging is included? The size of hysteresis? The height of the collapse…?

The methods are currently split in two parts but I would merge them. In particular, the equations are needed to understand the results. Also, the way adaptive foraging is modeled needs to be more explained (make figures, show some dynamics, give an intuition of how alpha varies depending on the relative abundance of species by building a few toy examples). Sharing the code would have greatly helped.

I give more detailed comments below.

Detailed comments

l. 2-4: “Previous studies…”: specify “previous theoretical studies….” As far as I know, there is no empirical observation of such behavior

l.5-7: “we show that…”: the way things are formulated, it seems that the possibility for rate-induced tipping points is a primary result of your work. But this is well-known and expected in model where you have bistability. Maybe it’s only a matter of formulation (things are written in terms that are too general lines 5-7), but I would try to more specifically highlight the novel results of the study, which as far as I understand are related to the effect of adaptive foraging on the dynamics of the model. These results are mentioned l. 8 (“additionally”), as something additional, whereas they should probably be the center of the abstract.

L. 9-11: Because of the various reasons explained in this text, I would delete l. 9-11 (from “Thus, ecosystem management…”

l. 13-18: In the same way as for the abstract, the author summary should highlight the novelty of the paper better. The way things are written, it seems that the results were already well-known in the literature.

l. 20: “pollinator communities, ecosystems of plant and pollinator species…”: the formulation is incorrect. Communities are not ecosystems… Maybe just reformulate as “Communities of plants and pollinators…”?

l. 60-61: the way things are written is clumsy. I agree that I don’t know of a theoretical study on plant-pollinator networks that specifically looked at rate-induced tipping. However, the theoretical results on the topic are so generic that we know that this can happen in such system and we don’t need to specifically test it here. What I find more interesting and novel here is the question of how adaptive foraging affect rate-induced tipping. I would try to reformulate this.

l. 78: A reference is needed at the end of the sentence “Adaptive networks…”

l. 88: “to test these hypothesis”: which ones? There is no explicit hypothesis formulated (as far as I could see).

l. 108: please move the equations and the rest of the methods here. It’s currently impossible for the reader to understand the results without having gone through the more detailed method. In particular, I did not understand the type of model you studied, and more specifically, how adaptive foraging was introduced in the model from the text alone. This point needs to be explained in more details.

L. 115: “when initial abundances are low”: does this mean that you ran simulations with high vs low initial abundances? How did initial conditions affect the outcome of the results and where is that shown?

Figure 4: Line 104, you wrote: "At q = 0, there is no resource competition and, thus, no adaptive foraging. For q > 0, there is adaptive foraging." So why don't we have the same results between the left and right panels for q=0?

I was confused at this point, but I think that’s also because I did not find the methods very clear on how adaptive foraging was incorporated in the model and how its effect was investigated. The fact that the methods is currently split in two parts and that the code for the analyses was not shared with the reviewers (or maybe I missed it?) did not help.

l. 254: I did not understand how you decide whether a connection exists? How do you generate the skeleton of the network?

In the equations after l. 258, I don’t understand what Kp and KA are: are they the average number of connections of plants and pollinators (if it’s “the number of connections a plant has”, as defined l. 256, then why doesn’t it have the index i or j of the concerned plant/pollinator species?

Equations after l. 256: there is no \\rho in equation 1 so it’s unclear exactly how to reconciliate those.

Equation 3, between l. 277 and 278: this is a key element of the model which is not stressed at all in the text. It would be important to highlight this point more (and earlier in the text), so that the reader understands that’s one of the originality of the paper.

l. 286: please replace “ecosystems” by “communities” (you’re not modeling ecosystems here)

l. 287: “…had a feasible solution,…” I found the formulation unclear: do you mean that had a feasible solution (even when some species may go extinct) or that had a solution such that all species initally presnet survive?

l. 291-292: “can be seen as a measure of resilience”. Not of resilience (or according to which definition of resilience). You could say persistence? Or more generally ‘stability’?

Reviewer #2: Please see the attached PDF for detailed comments. The paper is undoubtedly interesting and I recommend some minor revisions.

**Have the authors made all data and (if applicable) computational code underlying the findings in their manuscript fully available?**

Reviewer #1: None

Reviewer #2: Yes

PLOS authors have the option to publish the peer review history of their article (what does this mean?). If published, this will include your full peer review and any attached files.

Reviewer #1: No

Reviewer #2: No
---

## [Decision Letter · Decision Letter 1]

7 Nov 2023

Dear Dr. Vasconcelos,

Thank you very much for submitting your manuscript "Adaptive Foraging of Pollinators Fosters Gradual Tipping under Resource Competition and Rapid Environmental Change" for consideration at PLOS Computational Biology. As with all papers reviewed by the journal, your manuscript was reviewed by members of the editorial board and by two independent reviewers.

Based on the reviews, we are likely to accept this manuscript for publication, providing that you address the additional suggestions/questions of the referees. There is room for improvement in the writing in the transitions between sections and in removing repetitions throughout the text. 

Sincerely,

Mercedes Pascual

Academic Editor

PLOS Computational Biology

Natalia Komarova

Section Editor

PLOS Computational Biology

Reviewer's Responses to Questions

**Comments to the Authors:**

Reviewer #1: This paper has been extensively revised, and it is now much clearer in terms of focus and objectives. The novelty compared to previous studies is also better highlighted.

It is great to have included the model in the main text. Regarding the structure of the Method’s part, I would have one Model & Methods part but this is a personal preference (and I am not sure what is the typical structure for Plos Comp Biol papers).

One minor thing:

l. 60-61 : « However, most of the theoretical work on pollinator communities has found evidence for the potential for tipping points23,25–29, especially those based on Lotka-Volterra type of dynamics. »

I would say ‘some theoretical work’ because you cannot reduce the large number of theoretical studies of plant-pollinator models to these few ones studying tipping points.

Reviewer #2: Please see the attached PDF for more detailed comments. The revision has addressed the main issues and improved the paper considerably. I have some additional minor suggestions and questions.

**Have the authors made all data and (if applicable) computational code underlying the findings in their manuscript fully available?**

Reviewer #1: Yes

Reviewer #2: Yes

PLOS authors have the option to publish the peer review history of their article (what does this mean?). If published, this will include your full peer review and any attached files.

Reviewer #1: No

Reviewer #2: No

Figure Files:

Data Requirements:

Reproducibility:

References:

---

## [Editor Report · Decision Letter 2]

15 Dec 2023

Dear Dr. Vasconcelos,

We are pleased to inform you that your manuscript 'Adaptive Foraging of Pollinators Fosters Gradual Tipping under Resource Competition and Rapid Environmental Change' has been provisionally accepted for publication in PLOS Computational Biology.

Best regards,

Mercedes Pascual

Academic Editor

PLOS Computational Biology

Natalia Komarova

Section Editor

PLOS Computational Biology

---

## [Editor Report · Acceptance letter]

3 Jan 2024

PCOMPBIOL-D-23-00333R2 

Adaptive Foraging of Pollinators Fosters Gradual Tipping under Resource Competition and Rapid Environmental Change

Dear Dr Vasconcelos,

I am pleased to inform you that your manuscript has been formally accepted for publication in PLOS Computational Biology. Your manuscript is now with our production department and you will be notified of the publication date in due course.

With kind regards,

Zsofi Zombor
